# *PCYT1A* Missense Variant in Vizslas with Disproportionate Dwarfism

**DOI:** 10.3390/genes13122354

**Published:** 2022-12-13

**Authors:** Odette Ludwig-Peisker, Emily Ansel, Daniela Schweizer, Vidhya Jagannathan, Robert Loechel, Tosso Leeb

**Affiliations:** 1Institute of Genetics, Vetsuisse Faculty, University of Bern, 3001 Bern, Switzerland; 2BluePearl Pet Hospital, Raleigh, NC 27616, USA; 3Division of Clinical Radiology, Department of Clinical Veterinary Medicine, Vetsuisse Faculty, University of Bern, 3001 Bern, Switzerland; 4VetGen, Ann Arbor, MI 48108, USA

**Keywords:** dog, *Canis lupus familiaris*, skeleton, morphology, skeletal dysplasia, chondrodysplasia, animal model, precision medicine

## Abstract

Disproportionate dwarfism phenotypes represent a heterogeneous subset of skeletal dysplasias and have been described in many species including humans and dogs. In this study, we investigated Vizsla dogs that were affected by disproportionate dwarfism that we propose to designate as skeletal dysplasia 3 (SD3). The most striking skeletal changes comprised a marked shortening and deformation of the humerus and femur. An extended pedigree with six affected dogs suggested autosomal recessive inheritance. Combined linkage and homozygosity mapping localized a potential genetic defect to a ~4 Mb interval on chromosome 33. We sequenced the genome of an affected dog, and comparison with 926 control genomes revealed a single, private protein-changing variant in the critical interval, *PCYT1A*:XM_038583131.1:c.673T>C, predicted to cause an exchange of a highly conserved amino acid, XP_038439059.1:p.(Y225H). We observed perfect co-segregation of the genotypes with the phenotype in the studied family. When genotyping additional Vizslas, we encountered a single dog with disproportionate dwarfism that did not carry the mutant *PCYT1A* allele, which we hypothesize was due to heterogeneity. In the remaining 130 dogs, we observed perfect genotype–phenotype association, and none of the unaffected dogs were homozygous for the mutant *PCYT1A* allele. *PCYT1A* loss-of-function variants cause spondylometaphyseal dysplasia with cone–rod dystrophy (SMD-CRD) in humans. The skeletal changes in Vizslas were comparable to human patients. So far, no ocular phenotype has been recognized in dwarf Vizslas. We propose the *PCYT1A* missense variant as a candidate causative variant for SD3. Our data facilitate genetic testing of Vizslas to prevent the unintentional breeding of further affected puppies.

## 1. Introduction

Disproportionate dwarfism represents a subset of the skeletal dysplasias, a large and heterogeneous group of heritable disorders affecting the development and growth of chondro-osseous tissues [1]. In 2019, the Nosology Committee of the International Skeletal Dysplasia Society published a list of 461 different diseases in humans, which can be classified into 42 groups according to their clinical, radiographic and molecular phenotype. For 425 diseases, pathogenic variants have already been identified with the use of next-generation sequencing technologies [2]. Dwarfism can be categorized into proportionate and disproportionate dwarfism. Proportionate dwarfism is characterized by the trunk and limbs being proportionally small, while in disproportionate dwarfism only the limbs are shortened, while the trunk has a normal size [3].

Dwarfism may be a part of the breed standard in different domestic animal species and is characteristic for breeds such as Terriers [4], Munchkin cats [5], Dexter cattle [6] or Shetland ponies [7]. Dogs are known to represent the mammalian species with the greatest range of morphological diversity. Due to domestication and strong artificial selection, extensive variations in the size and shape of the skeletal bones exist between different dog breeds. For example, short-legged dogs, such as scent hounds, are typically purpose-bred dogs [4]. Chondrodysplasia, which can be a cause of disproportionate dwarfism, is a primary requirement for various dog breeds, such as Dachshund, Corgi or Basset Hound. In these dogs, the growth plates calcify prematurely during the growth process, which leads to shortened and curved bones. Insertions of *FGF4* retrogenes in different chromosomal locations were shown to cause chondrodysplasia and/or chondrodystrophy in at least 19 different dog breeds [8]. An insertion of an *FGF4* retrogene on chromosome 18 produces the chondrodysplasia phenotype without significantly increasing the risk of disc hernias [8,9]. Many short-legged breeds such as Dachshunds, French Bulldogs or Corgis have a high risk of intervertebral disc herniation. This increased risk is caused by insertion of an *FGF4* retrogene on chromosome 12, leading to chondrodystrophy, which comprises pathologic calcification of the intervertebral discs [9,10].

Both desired and undesired forms of disproportionate dwarfism can lead to health issues. There are various reports about undesired disproportionate dwarfism in different dog breeds. In the Dogo Argentino breed, a *PRKG2* splice site variant represents a candidate causal variant for disproportionate dwarfism characterized by forelimb deformations and a disproportionally large head [11]. A missense variant in *COL11A2* was reported in Labrador Retrievers with a mild form of disproportionate dwarfism, designated as skeletal dysplasia 2 (SD2) [12]. In Norwegian Elkhounds, a nonsense variant in *ITGA10* was described, and proposed to cause chondrodysplasia in nine affected dogs [13]. A 130 kb deletion including parts of the *SLC13A1* gene was detected in miniature poodles affected by osteochondrodysplasia [14]. Variants in *COL9A2* in Samoyeds and *COL9A3* in Labrador retrievers and Northern Inuit dogs were described to produce oculoskeletal dysplasia, a form of short-limbed dwarfism with severe ocular defects [15,16].

Dwarfism may also be caused by congenital hormone deficiency, which typically leads to proportionate dwarfism and a persistent puppy coat. Pituitary dwarfism, caused by an intronic deletion in the *LHX3* gene, was described in four genetically diverse dog breeds [17,18,19]. Another intronic variant in *POU1F1* was found in Karelian Bear dogs with pituitary dwarfism [20]. A single case of a Chihuahua with proportionate dwarfism, and episodes of hypoglycemia and collapse, had a 6 bp deletion in the *GH1* gene [21].

The present study was initiated after a veterinarian and breeder of Vizslas reported a litter in which two out of five puppies were affected by disproportionate dwarfism. Additional affected Vizsla puppies were recognized by other breeders. The aim of this study was to characterize the phenotype and identify the underlying genetic cause. We termed this phenotype skeletal dysplasia 3 (SD3).

## 2. Materials and Methods

### 2.1. Animals

This study included a total of 131 Vizslas (Appendix A). Based on their proportions, eight dogs were classified as affected by disproportionate dwarfism and 121 were classified as unaffected. Twenty-eight of the unaffected dogs were first-degree relatives of the cases and comprised eight parents (obligate carriers assuming autosomal recessive inheritance) and 20 full or half siblings. The other 95 unaffected Vizslas were more distantly related or of unknown relationship to the cases. Samples from one Vizsla family, comprising both parents, two affected and two unaffected puppies, were used for linkage analysis. A total of six related affected dogs were used for homozygosity mapping.

### 2.2. Clinical Examination

For this study, three affected and seven unaffected Vizslas were clinically examined with an emphasis on proportion and deformities of the body and limbs. Photographs and videos of four additional affected Vizslas were also evaluated. Shoulder height measurements were taken as described by the American Kennel Club [22] from 29 dogs, including all dogs that were clinically examined. Statistical analyses of height differences were performed using pairwise permutation tests using the infer package in R.

### 2.3. Diagnostic Imaging

Owners and veterinarians of affected and unaffected dogs were asked for existing radiographs, or to take radiographs of the radius and ulna and tibia/fibula in at least one mediolateral projection, and, if possible, a craniocaudal projection. As we included existing radiographs, this study was not conducted with a standardized protocol for the radiological examinations.

### 2.4. DNA Extraction, Linkage Analysis and Homozygosity Mapping

Genomic DNA was isolated with standard protocols from EDTA blood samples, cheek swabs or hair roots. The isolated genomic DNA was stored at −20 °C for further use.

Ten dogs were genotyped with the Illumina canine_HD SNV array comprising 220,853 markers (Neogen, Lincoln, NE, USA). Parametric linkage analysis with the genotyping data from the family was performed using PLINK software (v.1.90 Feb 2017). Variants with missing genotypes were excluded. Furthermore, markers with minor allele frequencies ≤ 0.01 and markers that were located on the sex chromosomes or contained Mendel errors were also excluded. A recessive disease model with full penetrance was used for parametric linkage analysis using Merlin software [23]. All regions with α = 1 were considered linked and potentially harboring the causative variant for this phenotype.

Homozygosity mapping was performed with six affected dogs using PLINK software (v1.90 Feb 2017) [24]. Markers with missing genotype data were excluded using the command --geno 0. Homozygous segments with shared haplotypes between all six cases were identified using the command --homozyg-group.

The linked and homozygous segments were compared using Excel spreadsheets. The overlapping regions were defined as critical intervals. The output of the linkage and homozygosity mapping is given in Appendix A.

### 2.5. Whole-Genome Sequencing

An Illumina TruSeq PCR-free DNA library with a ~380 bp insert size was prepared from a male affected dog (case 1). We collected 192 million 2 × 150 bp paired-end reads on a NovaSeq 6000 instrument (19.9× coverage). Mapping and alignment to the UU_Cfam_GSD_1.0 reference genome assembly were performed as described [25]. Briefly, in the first step, filtering for adaptors and trimming of the polyG in the read tails of the raw data were performed using *fastp* [26]. The alignment of the quality filtered reads to the reference genome UU_Cfam_GSD_1.0 was performed using *bwa* [27]. The sequence data were deposited in the European Nucleotide Archive under the study accession PRJEB16012 and the sample accession SAMEA110175539.

### 2.6. Variant Calling

Variant calling was performed using GATK HaplotypeCaller [28]. All unfiltered SNV and indel calls for the whole genome were listed in a gVCF file. SnpEff [29] software, together with NCBI annotation release 106 for the UU_Cfam_GSD_1.0 genome reference assembly, were used to predict the functional effects of the previously called variants. For variant filtering, we used 926 control genomes from dogs of different breeds (Appendix A). These genomes were all publicly available and accessions are listed in Appendix A.

### 2.7. Gene Analysis

Numbering within the canine *PCYT1A* gene corresponds to the NCBI RefSeq accession numbers XM_038583131.1 (mRNA) and XP_038439059.1 (protein).

### 2.8. PCR and Sanger Sequencing

PCR amplification followed by Sanger sequencing was used to genotype the *PCYT1A:*c.673T>C variant. To amplify the extracted genomic DNA AmpliTaqGold360Mastermix (Thermo Fisher Scientific, Waltham, MA, USA), together with the primers 5′-GAA TGT TGC TCC CAG TTT CC-3′ (Primer F) and 5′-CAG CAA AGA GGC ATT CAC TG-3′ (Primer R), were used. After an initial denaturation step for 10 min at 95 °C, 30 cycles of 30 s denaturation at 95 °C, 30 s annealing at 60 °C, and 1 min of polymerization at 72 °C followed. The PCR process finished with an extension step of 7 min at 72 °C, followed by cooling the samples to 4 °C for storage. Quality control of the PCR reaction was performed using a 5200 Fragment Analyzer (Agilent, Santa Clara, CA, USA). After treating the samples with exonuclease I and alkaline phosphatase, the sequencing reactions were performed. Sequencing reactions were purified using ethanol precipitation and analyzed on an ABI 3730 DNA Analyzer (Thermo Fisher Scientific, Waltham, MA, USA). The raw sequence data were analyzed with Sequencher 5.1 software (GeneCodes, Ann Arbor, MI, USA).

## 3. Results

### 3.1. Family Anamnesis and Clinical Examinations

All cases in this study were born to unaffected parents. Six of the eight cases could be traced back to common ancestors within five generations on their maternal and paternal lineages. The pedigree findings were consistent with an autosomal recessive mode of inheritance. For the remaining two cases we could not establish a relationship with the other cases (Figure 1).

At birth, affected puppies were not visibly distinguishable from their unaffected littermates. Deviation from the typical limb length and conformation became increasingly evident from three to five weeks of age. Affected puppies developed shorter and more bowed limbs than their littermates and a “knobby” appearance of the carpi.

As adults, affected individuals were shorter than their close relatives with shoulder heights of 42.5–50.8 cm. This was lower than the American Kennel Club breed standard, which specifies 55.9–61.0 cm (22–24 inches) for males and 53.3–58.4 cm (21–23 inches) for females. Seven affected dogs maintained a standard size and shape of the head and trunk. Case 8 was of similar height to the other cases, but additionally had mandibular brachygnathism (Appendix A; Appendix A).

On orthopedic examination, there was marked shortening of the brachium, an abducted elbow position and shortening and bowing of the antebrachium with thickening in the region of the metaphysis. However, significant carpal valgus or varus was not observed. Affected individuals maintained a wide-based stance in the forelimbs. The hindlimbs were similarly shortened, although in most of the cases they appeared slightly less affected than the forelimbs (Figure 2). At the time of writing, affected dogs were between two and seven years of age; none had systemic illnesses or vision problems diagnosed by their primary care veterinarians. One dog was evaluated by a veterinary ophthalmologist and ocular exam findings were normal except for early cataract formation (not suspected to be inherited).

### 3.2. Diagnostic Imaging

Radiographs from three affected (cases 4, 6, 8) and two unaffected dogs were available. The radiological findings in the affected dogs indicated variable severity of the skeletal changes.

In case 6, angular limb deformity with procurvatum of both humeri and procurvatum and varus deformity of both radii was noted (Figure 3a,b). Varus of both tibiae was also visible with marked thickening and shortening of long bones. In particular, the humeri and femora showed shortening, resulting in a shorter brachium in relation to the antebrachium and a shorter thigh in comparison to the lower leg (Figure 3c,d). The humerus and femur showed marked thickening of the epiphysis with a flattened articular surface (Figure 3c,d). In case 4, the thickening and shortening was less obvious, but the humerus, radius and ulna showed a procurvatum.

Case 8, for which we could not establish any relationship with the other affected dogs, showed similar limb deformities to the other cases. However, this dog additionally had a shortened lower jaw (Appendix A).

### 3.3. Genetic Analyses

Linkage analysis in a family with two affected and two unaffected offspring revealed 18 linked intervals, comprising 206 Mb with a maximum LOD score of 0.85, which were distributed over 15 different chromosomes. Homozygosity mapping in six affected dogs identified seven intervals with shared haplotypes, distributed over six chromosomes and totaling 9 Mb. The intersection of both analyses resulted in two closely neighboring segments of 1.4 Mb and 2.0 Mb on chromosome 33 that were considered critical intervals (Chr33:26,941,791-28,331,855 and Chr33:29,015,236-31,008,198).

We sequenced the genome of one of the affected dogs and searched for homozygous variants in the previously defined critical intervals. Private variants were identified by a comparison with 926 control genomes (Table 1 and Appendix A).

This analysis identified eight private protein-changing variants out of the 1193 private variants in the sequenced dog. Only one of these variants was located in the critical intervals. This variant affected the *PCYT1A* gene encoding phosphate cytidylyltransferase 1A, choline, which has also been termed choline-phosphate cytidylyltransferase-α. The genomic variant designation is Chr33:30,067,814:g.A>G (UU_Cfam_GSD_1.0 assembly). It represents a missense variant, XM_038583131.1:c.673T>C, predicted to cause an exchange of a highly conserved amino acid in the PCYT1A protein XP_038439059.1:p.(Y225H). We confirmed the presence of the *PCYT1A* missense variant using Sanger sequencing (Figure 4).

The genotypes at the variant showed perfect co-segregation with the phenotype in all available members of the sampled pedigree (Figure 1). Seven out of eight cases were homozygous for the mutant *PCYT1A* allele. Only case 8, without a known relation to the other cases, was homozygous for the wildtype allele. Seven of the eight obligate carriers carried the *PCYT1A* missense variant in a heterozygous state. Only the dam of case 8 had a homozygous wildtype genotype.

In a cohort of 95 unaffected Vizslas, 84 dogs were homozygous for the wildtype allele and 11 dogs carried the *PCYT1A* variant in a heterozygous state. This corresponds to a mutant allele frequency of 5.8% and a carrier frequency of 11.6% (Table 2 and Appendix A). As these dogs comprised several members of the extended pedigree, the allele and carrier frequencies probably represent upper boundaries of the values in the entire Vizsla population.

### 3.4. Genotype–Phenotype Correlation

We obtained shoulder height measurements from a total of 25 adult Vizslas. The homozygous mutant dogs were on average 11 cm shorter than the unaffected dogs. In a more detailed analysis considering sex-specific differences, homozygous mutant males and females were significantly shorter than heterozygous or homozygous wildtype dogs (Figure 5). This is consistent with a recessive mode of inheritance of the trait.

## 4. Discussion

In this study, we identified the *PCYT1A*:p.Y225H missense variant as a candidate causative variant for an autosomal recessive form of disproportionate dwarfism in Vizslas, which we propose to designate as skeletal dysplasia 3 (SD3). The phenotype is predominantly characterized by a shortening and deformation of the humerus and femur, but other long bones are also altered. There is some variability in the phenotype. We encountered one dog with disproportionate dwarfism (case 8) that phenotypically resembled the seven *PCYT1A* mutant cases although it did not carry the mutant allele. We assume that the phenotype of this dog was caused by another genetic variant and/or environmental factors. This assumption is supported by the observation that its phenotype showed some deviation from the other cases, notably a shortened lower jaw (Appendix A). Heterogeneity in skeletal dysplasias is well documented in humans and dogs [2,4,12].

This study provides only a very limited initial characterization of the SD3 phenotype. The oldest affected Vizsla in our study was seven years old at the time of writing and had no overt health problems. However, the extended hip radiographs of one affected dog (case 6) indicated hip dysplasia with secondary signs of osteoarthritis that might cause lameness. Future clinical studies are required to comprehensively characterize the phenotype and to clarify whether SD3 poses a significant risk for the health and wellbeing of affected dogs.

*PCYT1A* codes for choline-phosphate cytidylyltransferase-α (also known as CTP: phosphocholine cytidylyltransferase-α), an enzyme that regulates the biosynthesis of phosphatidylcholines [30]. Its function is the conversion of phosphocholine into cytidine diphosphate-choline [31]. Phosphatidylcholines belong to the phospholipids and are some of the most important components of eukaryotic membranes [32]. Phosphatidylcholines are composed of a glycerophosphoric acid with two variable fatty acids and one choline group. The enzyme PCYT1A is composed of four functional domains: N, C, M and P [33]. The N-terminal domain (N, ~75 residues) is followed by a catalytic domain (C, ~150 residues), a membrane binding domain (M, ~60 residues) and a phosphorylated tail (P, ~50 residues). Phosphate cytidylyltransferases exist in every domain of life including bacteria, plants and animals [33]. PCYT1A forms a homodimer in its soluble form. The activity of the enzyme itself is regulated by reversible interactions between the membrane and the amphipathic helical membrane binding domain (M) [34]. This amphipathic helix has a bend between the fifth α-helix (αE) and the M-helix [35]. The p.Y225H variant in affected Vizslas is located exactly at this bend and is predicted to exchange the neutral tyrosine with a positively charged histidine.

Phosphatidylcholines, the biosynthesis of which is catalyzed by PCYT1A, are not only part of cell membranes but also contribute to the formation of vesicle membranes [36]. Matrix vesicles are extracellular organelles and play a role in the mineralization of the endochondral bone tissue [37]. They are released by cells in the growth plate and attach to the extracellular matrix. It has been shown that their primary function in the mineralization process is providing sites for initial calcium phosphate crystal formation. Thus, we speculate that the mutant *PCYT1A* allele might interfere with the normal formation of matrix vesicles and mineralization of the endochondral bone tissue.

In humans, different variants in the *PCYT1A* gene may cause three discernible phenotypes: spondylometaphyseal dysplasia with cone–rod dystrophy (SMD-CRD) [31,38], lipodystrophy [39] and isolated retinal dystrophy [40]. Nine different missense or very late frameshift variants in *PCYT1A* were described to cause SMD-CRD, which phenotypically resembles the disproportionate dwarfism phenotype in affected Vizslas [31,38]. One of these variants, p.R223S, is located only two amino acids away from p.Y225H found in affected dogs [38]. The human patients all showed bowing of the long bones, platyspondyly, metaphyseal irregularity and cone–rod dystrophy.

In knockout mice, complete *Pcyt1a* deficiency leads to embryonic lethality. Homozygous *Pcyt1a^−/−^* embryos fail to form blastocysts and do not develop past embryonic day 3.5 [41]. It is currently not clear whether human patients with SMD-CRD and Vizslas with disproportionate dwarfism retain some residual enzymatic activity allowing them to successfully complete embryonic development, or whether the difference in phenotype between mice and humans or dogs is due to other factors.

Vision impairment seen in most, but not all human patients with *PCYT1A* variants [31,38,40] was not reported in any of the seven homozygous *PCYT1A* mutant dogs in our study. However, only one of these cases was evaluated by a veterinary ophthalmologist in search of signs of cone–rod dystrophy (at seven years of age). Further ophthalmological examinations including older dogs are needed to clarify whether homozygous *PCYT1A* mutant dogs are at increased risk of developing any ocular phenotypes.

We provide several lines of evidence supporting the causality of the *PCYT1A:*p.Y225H missense variant: Apart from case 8, which we consider to represent a different phenotype, we demonstrated perfect genotype–phenotype association in a moderately sized cohort of dogs as well as the correct co-segregation of the genotypes with the phenotypes in a large pedigree. *PCYT1A* loss-of-function variants in humans cause closely related phenotypes. Finally, the canine *PCYT1A:*p.(Y225H) variant affects an evolutionarily highly conserved amino acid residue. While this evidence does not conclusively prove the causality, we deem it sufficient to classify the variant as likely pathogenic according to human diagnostic criteria [42].

## 5. Conclusions

In conclusion, we describe a new form of recessively inherited disproportionate dwarfism in Vizslas and report the *PCYT1A:*p.Y225H missense variant as candidate causative variant for this phenotype, which we propose to designate as skeletal dysplasia 3 (SD3). Our data facilitate genetic testing of Vizslas to prevent the unintentional breeding of further affected puppies.

## Figures and Tables

**Figure 1 genes-13-02354-f001:**
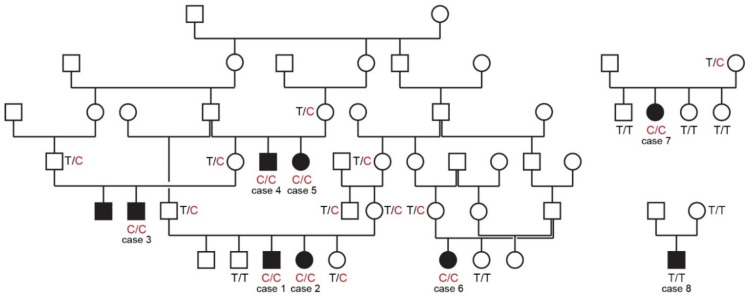
Pedigrees of affected Vizslas with disproportionate dwarfism. Circles represent females and squares males. Affected dogs are indicated by filled symbols. Six of the eight cases could be traced back to common ancestors in one large pedigree shown on the left side. Genotypes at the *PCYT1A:*c.673T>C variant are indicated for animals from which DNA samples were available.

**Figure 2 genes-13-02354-f002:**
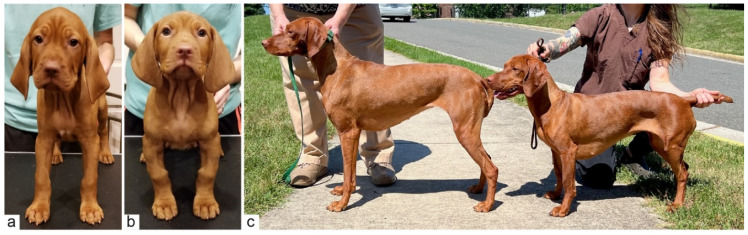
Clinical photos of affected and unaffected Vizslas. (**a**) Unaffected male control puppy at seven weeks of age. (**b**) Affected male littermate (case 1) of the dog shown in (**a**) at the same age. Note the deformity and shortening of both front limbs. (**c**) On the right, a 2-year-old affected female dog (case 6) next to its unaffected 3-year-old half-sister with standard proportions.

**Figure 3 genes-13-02354-f003:**
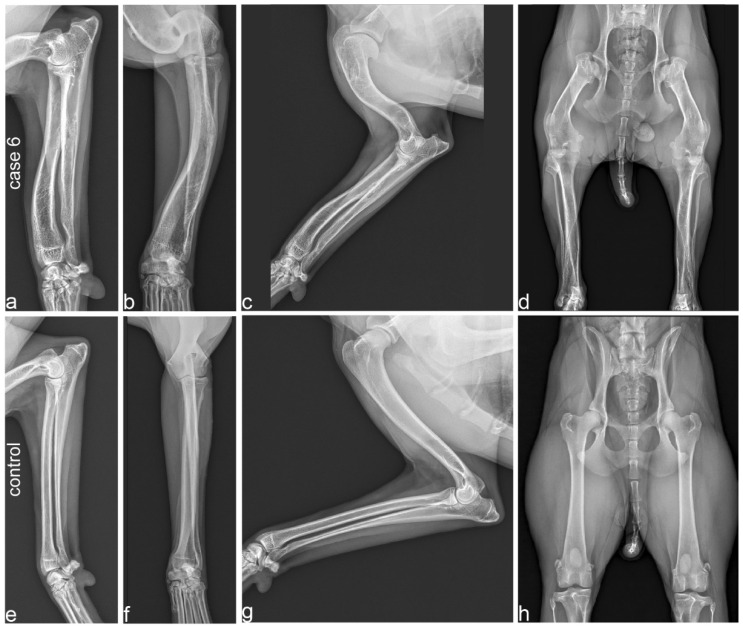
Radiographs of a 2-year-old affected Vizsla (case 6; **a**–**d**) and its 3-year-old unaffected half-sister (**e**–**h**). (**a**,**b**) Orthogonal radiographs of the radius and ulna in case 6 show procurvatum and varus deformity, and thick and short bones. (**c**) Humerus and antebrachium of the affected Vizsla. Note the marked deformity and shortening of the humerus, flattened articular surface and deformity of the humerus. (**d**) Extended hip radiograph of the affected dog. Note the marked shortening and deformity of the femur, including its epiphysis, resulting in subluxation of the hip joint and secondary deformity of the acetabula. (**e**,**f**) Orthogonal radiographs of the radius and ulna. All long bones of the unaffected control dog demonstrate a normal limb axis and normal relation between the length and thickness of the bones. (**g**) Humerus and antebrachium of the unaffected control. (**h**) Extended hip radiograph of the unaffected dog. Please note that the scale of the radiographs is not standardized.

**Figure 4 genes-13-02354-f004:**
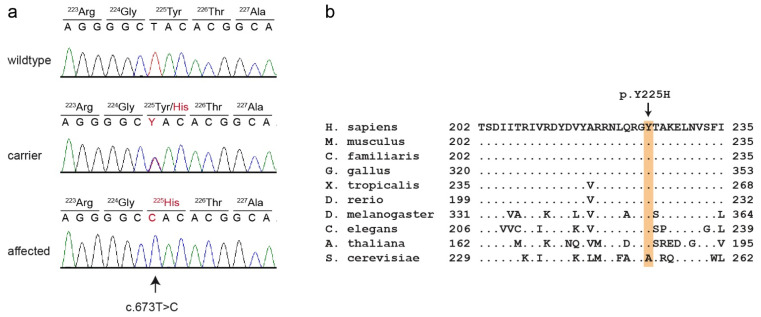
Details of the *PCYT1A*:c.673T>C variant. (**a**) Representative electropherograms of three dogs with different genotypes are shown. The variable position is indicated by an arrow. The altered base and the altered amino acid translation are indicated in red. (**b**) Multiple-species alignment of the *PCYT1A* amino acid sequences in the region harboring the p.Y225H variant. The variant affects a highly conserved tyrosine residue. Accession numbers: human (*Homo sapiens*) NP_001299602.1; mouse (*Mus musculus*) NP_034111.1; dog (*Canis lupus familiaris*) XP_038439059.1; chicken (*Gallus gallus*) XP_046754443.1; frog (*Xenopus tropicalis*) XP_031758151.1; zebrafish (*Danio rerio*) NP_001017634.1; fly (*Drosophila melanogaster*) NP_647621.1; worm (*Caenorhabditis elegans*) P49583.2; Arabidopsis (*Arabidopsis thaliana*) Q42555; yeast (*Saccharomyces cerevisiae*) NP_011718.1.

**Figure 5 genes-13-02354-f005:**
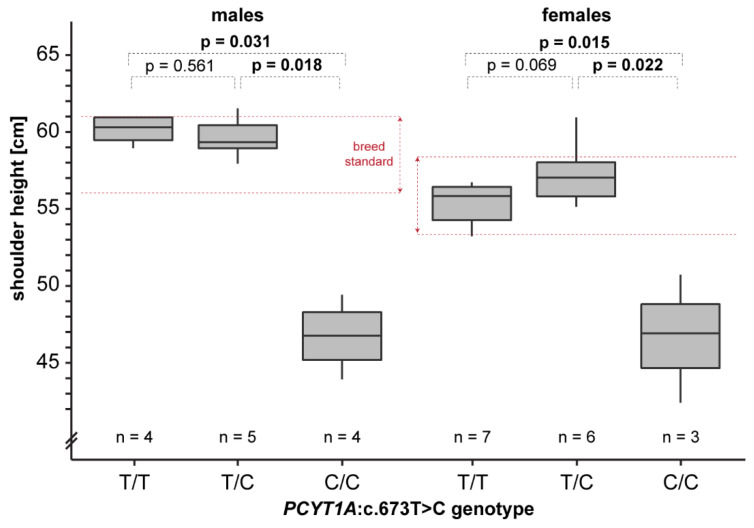
Genotype–phenotype correlation. Shoulder height measurements of Vizslas displayed according to their genotypes are given as box plots. Note that homozygous mutant dogs are markedly shorter than the breed standard. Pairwise permutation tests with 10,000 iterations were used to calculate *p*-values.

**Table 1 genes-13-02354-t001:** Variant filtering results of the whole genome sequence from the affected Vizsla against 926 control genomes.

Filtering Step	Homozygous Variants
Variants in the affected Vizsla	2,905,167
Private variants	1193
Private protein-changing variants	8
Private protein-changing variants in critical intervals	1

**Table 2 genes-13-02354-t002:** Association of the genotypes at the *PCYT1A*:c.673T>C variant with disproportionate dwarfism in the studied Vizslas.

Phenotype	T/T	T/C	C/C
Cases (*n* = 8)	1 ^1^	0	7
Obligate carriers (*n* = 8)	1 ^2^	7	0
Unaffected 1st degree relatives (*n* = 20)	12	8	0
Other controls (*n* = 95)	84	11	0

^1^ This dog, case 8, might be affected by a different form of disproportionate dwarfism (Appendix A). ^2^ This dog is the dam of case 8.

## Data Availability

The accessions for the sequence data reported in this study are listed in Appendix A.

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
