# Peer review of "PCYT1A Missense Variant in Vizslas with Disproportionate Dwarfism"

_genes, 2022, doi:10.3390/genes13122354_

Round 1

Reviewer 1 Report

As I expect from this laboratory, this is an excellent study and manuscript. For once, I have no criticism.  Perhaps I am getting too complacent.

Author Response

(1)

As I expect from this laboratory, this is an excellent study and manuscript. For once, I have no criticism.  Perhaps I am getting too complacent.

Response: Thank you very much for these very kind compliments!

Reviewer 2 Report

Review of Ludwig-Peisker et al

Lines 35-85 The introduction is very clearly written, comprehensive without being overly long.

Line 91 - maybe better to say it as “7 parents (obligate carriers under an autosomal recessive model of simple inheritance) and…”   

Also here and elsewhere, single digit numbers are written as numerals instead of words “7” not “seven” etc. and typically numbers below ten are written out in words. It may be a specific style choice, but anyway something to double check and make sure you intended to do.

172 – fig 1 legend    "animals, from" which   …prob should remove the comma.

224-240 – very nice story with the intersection and boiling everything down to the critical interval at one extended locus. Remarkable given the small number of animals involved.

244 – Fig 4a, very clean Sanger sequencing, very nice result. Fig 4b, wow the entire polypeptide segment shown is identical aa sequence in mammals, high sequence conservation. Y225 site is also highly conserved across eukaryotes.

265 – Table 2 is very compelling. Of course it would be a tidy story if the one affected dog not C/C was also homozygous for the C allele, but that doesn’t detract from the story. Genetics is complicated, the locus (at least within this one breed’s genetic background context) appears to be ~single locus “simple” with high penetrance. Additionally, the 8th affected dog has a distinct collection of phenotypes (line 181). I see the authors as reporting all of this well.

Line 272 – remove the phrase “suggestive significance”. Frequentist stats significance is present or absent, no in-between. At an alpha criterion of 0.05 you don’t have a significant difference between these two genotype categories. But that’s fine, the trend is obvious and matches the males. It’s clearly the tiny sample sizes, as you say. The result is strong, about as strong as it could be given the small samples.

Line ~270 again – I don’t see where you state the statistical test used for these pairwise differences. Wilcox rank sum? Randomization test? Either would be appropriate though I’d think randomization would have more power than rank-sum. T test (or one-way ANOVA of the 3 genotypes) is probably not appropriate given the tiny sample sizes and resulting inability to assess normal distribution in the samples.

275 – fig 5 is fine as is (no need to remake the figure), but with such small numbers of n per category it would have been sensible to just do dot plots or perhaps jittered dots within boxplots if the boxes were really wanted. I do appreciate the authors putting the n=3, 4, 6 right on the face of the plot.

279 – as [the/a] causative…

Discussion is very good

Line 356 – XXX… need to replace with a URL perhaps.

Overall, this is a really excellent report: clear results, clear writing and display items. With only minor edits this should be published. The authors have done fine work.

Author Response

(1)

Lines 35-85 The introduction is very clearly written, comprehensive without being overly long.

Response: Thank you for the compliment.

(2)

Line 91 - maybe better to say it as “7 parents (obligate carriers under an autosomal recessive model of simple inheritance) and…”  

Response: We added the assumption of autosomal recessive inheritance to line 91.

(3)

Also here and elsewhere, single digit numbers are written as numerals instead of words “7” not “seven” etc. and typically numbers below ten are written out in words. It may be a specific style choice, but anyway something to double check and make sure you intended to do.

Response: We revised this according to the reviewer's comment. We ask that this is double-checked by the publisher during the production stage of the article, so that it is consistent with journal style.

(4)

172 – fig 1 legend    "animals, from" which   …prob should remove the comma.

Response: Revised accordingly.

(5)

224-240 – very nice story with the intersection and boiling everything down to the critical interval at one extended locus. Remarkable given the small number of animals involved.

Response: Thank you for the compliment.

(6)

244 – Fig 4a, very clean Sanger sequencing, very nice result. Fig 4b, wow the entire polypeptide segment shown is identical aa sequence in mammals, high sequence conservation. Y225 site is also highly conserved across eukaryotes.

Response: No change requested.

(7)

265 – Table 2 is very compelling. Of course it would be a tidy story if the one affected dog not C/C was also homozygous for the C allele, but that doesn’t detract from the story. Genetics is complicated, the locus (at least within this one breed’s genetic background context) appears to be ~single locus “simple” with high penetrance. Additionally, the 8th affected dog has a distinct collection of phenotypes (line 181). I see the authors as reporting all of this well.

Response: Thank you for the compliment.

(8)

Line 272 – remove the phrase “suggestive significance”. Frequentist stats significance is present or absent, no in-between. At an alpha criterion of 0.05 you don’t have a significant difference between these two genotype categories. But that’s fine, the trend is obvious and matches the males. It’s clearly the tiny sample sizes, as you say. The result is strong, about as strong as it could be given the small samples.

Response: Since the original manuscript submission, we were able to collect four additional height measurements from unaffected controls. This increase in sample size was sufficient to achieve significance as expected for a recessive trait. We revised the text and the figure accordingly.

(9)

Line ~270 again – I don’t see where you state the statistical test used for these pairwise differences. Wilcox rank sum? Randomization test? Either would be appropriate though I’d think randomization would have more power than rank-sum. T test (or one-way ANOVA of the 3 genotypes) is probably not appropriate given the tiny sample sizes and resulting inability to assess normal distribution in the samples.

Response: Thank you for this comment. We had originally used a student's t-test. Following the suggestion from the reviewer, we now use a permutation test (~randomization test). We revised figure 5 and its legend accordingly.

(10)

275 – fig 5 is fine as is (no need to remake the figure), but with such small numbers of n per category it would have been sensible to just do dot plots or perhaps jittered dots within boxplots if the boxes were really wanted. I do appreciate the authors putting the n=3, 4, 6 right on the face of the plot.

Response: We agree with the reviewer and will consider this in the future. As the reviewer stated that no change of the figures is necessary, we left the design of the figure unchanged.

(11)

279 – as [the/a] causative…

Response: We added an "a".

(12)

Discussion is very good

Response: Thank you for the compliment.

(13)

Line 356 – XXX… need to replace with a URL perhaps.

Response: We assume that the missing URL for the supplementary data will be added by the publisher during the production state.

(14)

Overall, this is a really excellent report: clear results, clear writing and display items. With only minor edits this should be published. The authors have done fine work.

Response: Thank you very much for these very kind words!

Additional changes:

In the meantime, we obtained a few additional genotypes, height measurements and some additional information on first-degree relationships between dogs. We included the new data into the manuscript, which increased the total number of animals to 131 and led to minor changes in Figures 1 & 5 and Table 2. The additional data added further support to the main conclusions of the manuscript, which did not change with respect to the original submission.

Reviewer 3 Report

Well written.

1. English grammar and spelling mistakes should be corrected.

2. Methods should be written in detail.

3. Filteration steps should be written, and a flowsheet diagram should be added so readers can easily understand the flow or manuscript.

4. Future benefits of this research?

Author Response

(1)

English grammar and spelling mistakes should be corrected.

Response: We revised every specific language error pointed out by reviewer 2 and a few other typos.

(2)

Methods should be written in detail.

Response: With all due respect to the reviewer, this comment is vague and unclear. No specific request for changes is indicated. We think that our methods description is consistent with currently accepted standards in veterinary genetics. We did not change the manuscript with respect to this comment.

(3)

Filteration steps should be written, and a flowsheet diagram should be added so readers can easily understand the flow or manuscript.

Response: We added a supplementary figure illustrating the workflow for the genetic analysis (new Figure S1).

(4)

Future benefits of this research?

Response: This is clearly stated at the end of the abstract and the end of the conclusions. ("Our data facilitate genetic testing of Vizslas to prevent the unintentional breeding of further affected puppies.")

Additional changes:

In the meantime, we obtained a few additional genotypes, height measurements and some additional information on first-degree relationships between dogs. We included the new data into the manuscript, which increased the total number of animals to 131 and led to minor changes in Figures 1 & 5 and Table 2. The additional data added further support to the main conclusions of the manuscript, which did not change with respect to the original submission.